# Metabolite Profiling and Distribution of Militarine in Rats Using UPLC-Q-TOF-MS/MS

**DOI:** 10.3390/molecules25051082

**Published:** 2020-02-28

**Authors:** Limin Li, Bin Hao, Yulong Zhang, Shen Ji, Guixin Chou

**Affiliations:** 1Institute of Chinese Materia Medica, Shanghai University of Traditional Chinese Medicine, Shanghai 201203, China; lilimin_sifdc@163.com; 2Shanghai Institute for Food and Drug Control, Shanghai 201203, China; jishen2013@163.com; 3School of Pharmacy, Shanghai JiaoTong University, Shanghai 200240, China; bhao@sjtu.edu.cn (B.H.); yulong_zhang@sjtu.edu.cn (Y.Z.)

**Keywords:** militarine, metabolite identification, metabolic profiles, UPLC-Q-TOF-MS/MS

## Abstract

Militarine, a natural glucosyloxybenzyl 2-isobutylmalate, isolated from *Bletilla striata*, was reported with a prominent neuroprotective effect recently. The limited information on the metabolism of militarine impedes comprehension of its biological actions and pharmacology. This study aimed to investigate the metabolite profile and the distribution of militarine in vivo, which help to clarify the action mechanism further. A total of 71 metabolites (57 new metabolites) in rats were identified with a systematic method by ultra-high-performance liquid chromatography combined with quadrupole time-of-flight tandem mass spectrometry (UPLC-Q-TOF-MS/MS). The proposed metabolic pathways of militarine include hydrolyzation, oxidation, glycosylation, esterification, sulfation, glucuronidation and glycine conjugation. Militarine and its metabolites were distributed extensively in the treated rats. Notably, six metabolites of militarine were identified in cerebrospinal fluid (CSF), which were highly consistent with the metabolites after oral administration of gastrodin in rats. Among the metabolites in CSF, five of them were not reported before. It is the first systematic metabolic study of militarine in vivo, which is very helpful for better comprehension of the functions and the central nervous system (CNS) bioactivities of militarine. The findings will also provide an essential reference for the metabolism of other glucosylated benzyl esters of succinic, malic, tartaric and citric acids.

## 1. Introduction

Militarine, a glucosyloxybenzyl 2-isobutylmalate compound (Figure 1), is a major chemical constituent of the tuber of *Bletilla striata (Thunb.) Reichb.f.* [1,2]. *Bletilla striata* has been traditionally utilized for the treatment of traumatic bleeding, hemoptysis and hematemesis [3]. Recently, several studies have proved that militarine exhibits an effect on the central nervous system (CNS), which is involved in neuroprotection, learning and memory improvement, and intelligence improvement [4,5]. However, the blood–brain barrier (BBB) was the major obstacle preventing drug delivery to brain tissue, particularly for large-molecular-weight compounds and water-soluble molecules [6,7]. The high molecular weight of militarine (726 Da) makes it a big challenge to penetrate the BBB. One study in vitro proved that the main metabolic pathway of militarine in liver microsomes was the hydrolysis of ester bonds [8]. Sun et al. reported that militarine was rapidly transformed into gastrodin, α-isobutyl-malic acid and gymnoside I in rats after oral administration [9]. Gastrodin is well known as the main functional component of Chinese herbal medicine, *Gastrodia elata*. Gastrodin and its aglycone p-hydroxybenzyl alcohol could improve anxiety and depression, which have effects on Alzheimer’s disease and Parkinson’s disease [10,11,12,13,14,15,16]. Thus, gastrodin may be the efficient active metabolite of militarine on the CNS effect. Interestingly, we surprisingly found that compounds (Figure 1) with glucosylated benzyl esters’ structure, such as dactylorhin B, loroglossin, dactylorhin A and parishin C, also had an obvious protective effect on the CNS [17,18,19,20,21]. We speculate the central nervous affection of militarine and other glucosylated benzyl esters may be related to the gastrodin unit in the molecules.

Therefore, illuminating the metabolites and metabolic pathways of militarine would help to explain its pharmacological mechanisms. At present, only a few studies have reported the biotransformation of militarine in vivo or in vitro, in which four metabolites (gastrodin, p-hydroxybenzyl alcohol, gymnoside I, and isobutyl-malic acid) were identified [8,9,22]. We would perform integrative studies to determine the metabolites and distribution of militarine in rats, especially in the CSF and brain. Thus, in this study, we proposed a strategy to study the biotransformation of militarine in rats with ultra-high-performance liquid chromatography combined with quadrupole time-of-flight tandem mass spectrometry (UPLC-Q-TOF-MS/MS). A novel systematic data processing method, multiple characteristic fragments search, was developed to identify the structures of the metabolites. The metabolites in biological samples (plasma, bile, urine, feces, CSF, liver, kidney, spleen, lung, brain, heart and duodenum) were identified after rats were orally administrated with militarine, and then the metabolic pathways of militarine was described. Additionally, the study of metabolites in CSF after rats were orally administrated with gastrodin was performed to verify the possible bioactive metabolites of militarine, which could pass through the BBB then distribute into the brain. This is the first study to illustrate that the metabolites of militarine could be the primary functional compounds, which give a better comprehension of the CNS function pharmacological mechanism of militarine. This work provides a systematic study on the metabolism of militarine, which also could contribute to understanding the metabolite profiling of the other compounds with the same glucosylated benzyl esters’ structure.

## 2. Results and Discussion

### 2.1. Method Optimization

For detecting the trace metabolites of militarine in biological samples, solid-phase extraction (SPE) was used for sample preparation with higher detection sensitivity. The short-gradient UPLC run would result in the co-elution metabolites missing the isomers identification, like gymnoside II, which was undetected in a previous study [9]. Thus, a long ACQUITY BEH C_18_ column (2.1 × 100 mm, 1.7 μm) and suitable gradient were used to improve the separation of many isomers. The addition of formic acid in gradient elution improved the MS response of most metabolites, but reduced the detection sensitivity of gastrodin and p-hydroxybenzyl alcohol. In contrast, the addition of acetic acid could improve the sensitivity of gastrodin and p-hydroxybenzyl alcohol [23] but lead to peak broadening of isobutyl-malic acid. Thus, the final mobile phase was composed of 0.1% acetic acid in water and 0.5% formic acid in acetonitrile. The positive and negative ion modes were compared to find the best experimental conditions and to investigate the mass spectral behavior of the militarine and its potential metabolites. Some metabolites, such as isobutyl-malic acid, p-hydroxybenzoic acid and gastrodin-COOH (the alcohol hydroxyl of gastrodin oxidized to acid) performed poorly in positive ion mode, so the negative ion mode was selected for detecting militarine and its all metabolites.

### 2.2. The Strategy for Identification of Metabolites

To identify as many metabolites as possible, we developed an efficient approach based on UPLC-Q-TOF-MS/MS analysis coupled with multiple data processing techniques (Figure 2). Firstly, it is necessary to interpret the fragmentation patterns of militarine fully. As shown in Figure 3, the fragment ion at *m*/*z* 457.1690 was derived from the ion at *m*/*z* 725.2635 of militarine by the loss of a gastrodin unit (-C_13_H_16_O_6_, −268 Da), which also corresponds to the molecular ion of gymnoside I or gymnoside II. The fragment ions at *m*/*z* 285.0965 and 189.0758 were produced by further cleavage of militarine or the ion at *m*/*z* 457.1690, corresponding to gastrodin and isobutyl-malic acid, respectively. The fragment ion at *m*/*z* 161.0449 shows the glucose unit of militarine. Gastrodin produced the fragment ions at *m*/*z* 123.0440 and 105.0345 by the elimination of glucosyl and glucose. Five characteristic fragment ions at *m*/*z* 171.065, 153.0548, 127.0755, 129.0544 and 99.0799 were all from the ion at *m*/*z* 189.0758 as the existence of the isobutyl-malic acid unit.

Then, an IDA (information-dependent acquisition) analysis with dynamic background subtraction and mass defect filter (MDF) were performed in negative ion mode with full-scan datasets obtained. For the subsequent data mining processing, MetabolitePilot^TM^ 2.03 software (AB SCIEX, Framinghamcity, MA, USA, 2019) with MDF, product ion filter (PIF) and neutral loss filter (NLF) were utilized to screen possible metabolites of militarine in biosamples based on the accurate mass measurement and common biotransformation reactions. LC-MS/MS data were converted to mzXML format using ProteoWizard 3.0 software [24]. Then, multiple fragments search with MS-Dial 4.0 [25] (http://prime.psc.riken.jp/Metabolomics_Software/MS-DIAL/) was applied to supplement and refine possible metabolites based on characteristic fragment ions from the pattern of militarine, as well as the predicted metabolites by MetabolitePilot^TM^. The list of predicted fragment ions not only includes ions generated from the parent structure but also was extended by possible biotransformations for more comprehensive and reliable metabolite identification. The common fragments display a similar structure, thus identifying the putative metabolites based on the precursor information. The custom fragment ions and neutral loss molecules were listed in Appendix A. As a result, a rapid structural illumination of militarine homologous metabolites was implemented, and the metabolic pathways of militarine were proposed.

### 2.3. Identification of Metabolites

A total of 71 metabolites of militarine were tentatively identified in biological samples by the above-developed strategy. Information regarding these metabolites (retention times, formula proposed, product ions, metabolic process) is shown in Table 1. The proposed chemical structures of the metabolites are shown in Figure 4. Moreover, representative MS/MS spectra of all metabolites of militarine are shown in the Appendix A.

#### 2.3.1. Hydrolysis Metabolites

M51 and M56 showed the same [M − H]^−^ at *m*/*z* 457.17, corresponding to the molecular formula of C_21_H_30_O_11_, eluting at 12.7 and 13.0 min, respectively. The mass weight was 268 Da less than that of militarine, which suggested that the pair of isomers were produced by the loss of gastrodin units from militarine. The MS^2^ spectrum also showed the same characteristic fragment ions as that of militarine. By comparison with the authentic standards, M51 and M56 were identified as gymnoside II and gymnoside I, respectively.

M12 showed ions at *m*/*z* 345.1189 ([M + CH_3_COOH]^−^) and 285.0978 ([M − H]^−^), the molecular formula was calculated to be C_13_H_18_O_7_. Its MS^2^ spectrum showed the characteristic fragment ions at *m*/*z* 123.0449 (C_7_H_7_O_2_) and 105.0344 (C_7_H_5_O). Compared with the reference standard, M12 was further identified as gastrodin. M12 was also detected as a significant metabolite of parishin in rats [26].

M19 at 3.43 min showed [M − H]^−^ at *m*/*z* 123.0451, which is 162.05 Da less than M12. In the MS^2^ spectrum, the precursor produced the fragment ions at *m*/*z* 105.0351 and 77.0400 by sequential loss of H_2_O and CO. Therefore, M19 was identified as p-hydroxybenzyl alcohol and further validated by comparing the retention behavior and MS spectra with the reference standard.

M32 at 8.48 min showed an intense precursor ion at *m*/*z* 189.0766 ([M − H]^−^). By comparing the major fragments (including *m*/*z* 99.0815, 127.0768, 129.0565, and 171.0681) with those of militarine, M32 was identified as isobutyl-malic acid, which agreed with the previous studies [9].

#### 2.3.2. Oxidative Metabolites

M14 at 2.79 min showed [M − H]^−^ at *m*/*z* 299.0771, which was 14 Da higher than M12 (gastrodin). The fragment ions at *m*/*z* 137.0232 and 93.0334 suggested the presence of the p-hydroxybenzoic acid unit, while the precursor produced the fragment ion at *m*/*z* 137.0248 by the neutral loss of a glucose unit (−162.05 Da). Therefore, M14 was assigned as the oxidation product as gastrodin-COOH (the alcohol hydroxyl group of gastrodin was oxidized to carboxyl), which agreed with the previous studies [26,27].

M23 at 4.41 min showed an ion at *m*/*z* 329.0878 ([M + HCOO]^−^), with the calculated molecular formula of C_13_H_16_O_7_. There were two hydrogens less than M12. The fragment ion at *m*/*z* 121.0292 was consistent with the unit of p-hydroxy-benzaldehyde (M28), indicating the loss of a glucose unit (−162.05 Da) from the precursor ion at *m*/*z* 283.0823. M23 was assigned as a glycoside of p-hydroxybenzaldehyde; that is, the hydroxy group of gastrodin was oxidized to an aldehyde group [27].

M27 showed a precursor ion ([M − H]^−^) of *m*/*z* 137.0244 at 4.98 min, which was 162.05 Da less than M14, corresponding to the loss of a glucose unit. In the MS^2^ spectrum, the precursor ion formed the characteristic fragment ion at *m*/*z* 93.0340 by the elimination of COOH. Compared to the spectrum of the reference, M27 was assigned as p-hydroxybenzoic acid [26,27].

M28 at 6.43 min showed a precursor ion ([M − H]^−^) at *m*/*z* 121.0295, which was 162.05 Da less than M23, corresponding to the loss of a glucose unit. The precursor produced the characteristic fragment ion at *m*/*z* 92.0265 by the elimination of CHO. Furthermore, compared with the reference spectrum, M28 was assigned as p-hydroxybenzaldehyde [27].

M10 and M15 at 2.64 and 2.81 min gave the same precursor ion at *m*/*z* 301.09, which were an oxygen atom (16 Da) more than gastrodin (M12). The product ion at *m*/*z* 139.04 was formed by the loss of the glucose unit (162.05 Da), indicating the oxidation happed on the phenyl of the p-hydroxybenzyl alcohol group (*m*/*z* 123.05 + 16 Da). The phenolic hydroxyl substitution of the phenyl ortho and meta positions formed a pair of isomers.

M62 and M63 at 14.66 and 14.75 min showed the same precursor ion at *m*/*z* 741.26, which was 16 Da more than militarine. In the MS^2^ spectrum, the fragment ions at *m*/*z* 285.10 and 301.09, *m*/*z* 457.17 and 473.17 appeared simultaneously. Similar to M10 and M15, M62 and M63 were the hydroxylated metabolites of militarine.

#### 2.3.3. Glycosylation Metabolites

M29–M31 were eluted at 7.12, 7.42 and 7.84 min, respectively, which showed the same precursor ion ([M − H]^−^) at *m*/*z* 351.13. The molecular formula was calculated as C_14_H_24_O_10_, which was 162 Da (C_6_H_10_O_5_) higher than M32 (isobutyl-malic acid). M29 and M30 underwent the same fragmentation, showing strong responses at *m*/*z* 59.01 and 89.02, and weak response at *m*/*z* 127.08. However, M31 had an opposite result with the strong response at *m*/*z* 127.08 and the weak responses of the other two ions. Furthermore, the fragment ion at *m*/*z* 189.0 appeared only in M29 and M30, which suggested the glucosyl group was attached to the two carboxyl groups of isobutyl-malic acid, respectively. The glucose group of M31 was linked to the hydroxyl group of the isobutyl malate molecule, which was identified as dactylorhin C [28].

M34, M36, M39, M42, M43, M45, M47, M50 and M54 were eluted at 10.98, 11.30, 11.53, 11.82, 11.98, 12.29, 12.49, 12.68 and 12.96 min, respectively, which showed the same [M − H]^−^ at *m*/*z* 619.22 calculated as C_27_H_40_O_16_. The molecule weight was 162 Da higher than that of gymnoside II (M51) or gymnoside I (M56). The MS^2^ spectrum of M34 and M36 gave the characteristic fragment ions at *m*/*z* 439.16 and 351.13, which yielded correspondingly by the loss of glucose (180 Da) and gastrodin unit (268 Da) from the precursor ions. Similar to M31, the fragment at *m*/*z* 351.13 indicated that a glucosyl group was conjugated to the alcoholic hydroxyl group of isobutyl-malic acid section in the molecule of the gymnoside I (II) (gymnoside I or gymnoside II). The other metabolites produce the different characteristic fragment ions at *m*/*z* 447.15 and 323.10. The fragment ion at *m*/*z* 447.15 was formed due to the loss of the isobutyl-malic acid unit and the ion at *m*/*z* 323.10 showed the disaccharide group. So, the structure of other metabolites was assigned as a series of isomers of the glycosylation on the glucose molecule of gymnoside I (II).

M35, M38, M41, M44, M46 and M49 with same [M − H]^−^ at *m*/*z* 781.28 were eluted at 11.03, 11.34, 11.7, 12.13, 12.36 and 12.67 min, respectively, 162 Da higher than the metabolites with the fragment ion at *m*/*z* 619.22 (metabolites mentioned above). The MS^2^ spectra of these metabolites showed the disaccharide fragment ion at *m*/*z* 323.10 and the trisaccharide fragment ion at *m*/*z* 485.15 at the same time. Additionally, the precursor produced the fragment ion at *m*/*z* 609.20 by loss of the isobutyl-malic acid unit. Therefore, these compounds would be formed by the diglycosylation on the glucose molecule of gymnoside I (II).

M64 at 14.79 min and M65 at 14.92 min showed the same precursor ion at *m*/*z* 887.32, which was 162 Da higher than militarine. In the MS^2^ spectra, the disaccharide fragment ion at *m*/*z* 323.10 confirmed M64 and M65 as glucose-conjugated products of militarine.

#### 2.3.4. Glycine-Conjugated Metabolite

M21 at 3.83 min showed the precursor ion at *m*/*z* 194.0458, which was 57.0214 Da higher than M27, suggesting that it might be a glycine conjugate (NHCH_2_CO) of M27. After losing the carboxyl group, the fragment ion at *m*/*z* 93.0338 was consistent with the dominant fragment ion of M27.

#### 2.3.5. Sulfate Conjugated Metabolites

The metabolites conjugated with sulfate could show the characteristic fragment ions at *m*/*z* 79.96 and 96.96, indicating the presence of SO_3_^−^ and SO_4_^−^ in the chemical structure. Based on the ion mass difference between the deprotonated molecule and other product ions, the sulfate of the phase I metabolites with hydroxyl could be tentatively identified.

M1 and M11 with the same precursor ion at *m*/*z* 203.00 were 79.96 Da higher than M19 ([M − H]^−^ at *m*/*z* 123.05), indicating the presence of a sulfate unit. The typical fragment ions at *m*/*z* 123.05, 105.03 and 77.04 were the same as those of M19. Both the alcoholic and the phenolic hydroxyl groups of p-hydroxybenzyl alcohol could link with the sulfate unit. Then, M1 and M11 were tentatively identified as the sulfate of p-hydroxybenzyl alcohol. M11 appeared in the metabolites of parishin detected in rat plasma and urine [26].

M6 at 2.12 min and M8 at 2.26 min showed the precursor ions at *m*/*z* 218.99. The primary product ion at *m*/*z* 139.04 was 16 Da higher than M19 (p-hydroxybenzyl alcohol), which produced fragment at *m*/*z* 121.03 by the loss of H_2_O. Thus, M6 and M8 were tentatively identified as the sulfates of the hydroxylated p-hydroxybenzyl alcohol.

M9 at 2.61 min and M18 at 3.32 min showed the same precursor ions at *m*/*z* 365.06, suggesting that the molecular formula was C_13_H_18_O_10_S. By comparing their formula with that of M12 (gastrodin), M9 and M18 were identified as the sulfate conjugate of gastrodin. The product ion at *m*/*z* 241.00 in the MS^2^ spectrums was equivalent to the sum of glucose and SO_3_^−^, suggesting the conjugate site was located on the glucose unit of gastrodin.

M13 at 2.70 min and M25 at 4.68 min showed the same [M − H]^−^ at *m*/*z* 379.03, calculated molecular formula as C_13_H_16_O_11_S. The characteristic ions at *m*/*z* 299.08 and 137.02 were identical with the corresponding ions of M14, while the product ions at *m*/*z* 79.96, 96.96 and 241.00 indicated the presence of sulfate conjugated to the glucose group of M14. Thus, M13 and M25 were tentatively identified as sulfate conjugated metabolites of M14.

M20 at 3.53 min showed [M − H]^−^ at *m*/*z* 216.9812 and yielded fragment ions at *m*/*z* 137.0240 and 93.0342. The characteristic fragment ions were produced by the loss of SO_3_^−^ (216.9812→137.0240) and further loss of COOH (137.0240→93.0342), which indicated that M20 might be the sulfate conjugate of p-hydroxybenzoic acid.

M26 at 4.80 min showed the precursor ion at *m*/*z* 200.9862. The fragment ions were formed by the loss of SO_3_^−^ (200.9862→121.0288) and further loss of CHO (121.0288→92.0263), which indicated that M26 might be the sulfate conjugate of p-hydroxybenzaldehyde.

M57–M60 were eluted at 13.07, 13.37, 13.55, and 13.83 min, respectively. The four metabolites shared the same formula of C_21_H_30_O_14_S with [M − H]^−^ at *m*/*z* 537.13, which was 80 Da higher than gymnoside I (II). The characteristic fragment ions at *m*/*z* 129.06 and 189.07 were also the same as those of M51 or M56. Therefore, the four metabolites were tentatively identified as sulfate conjugated metabolites of gymnoside I (II).

#### 2.3.6. Glucuronide Conjugated Metabolites

To determine the glucuronide conjugated metabolites, the fragment ion at *m*/*z* 175.03 was one of the characteristics. Meanwhile, the ion mass difference (176.03 Da) from precursor ions and the characteristic fragment ions derived from the corresponsive phase I metabolite were other evidence. Additionally, the product ion at *m*/*z* 337.08 indicated that a glucuronide bonded to a glucose unit for some metabolites. A total of 18 glucuronide conjugated metabolites were found in the biological samples of rats after the oral administration of militarine.

M2 at 1.68 min and M24 at 4.42 min showed the same precursor ion at *m*/*z* 475.11, identified as glucuronide conjugated metabolite of M14. In the MS^2^ spectrum of M2, the fragment ions at *m*/*z* 93.03, 137.02 and 299.08 were agreement with M14. It meant the glucuronic acid was bound to the carboxyl group of M14. In the MS^2^ spectrum of M24, the product ion at *m*/*z* 337.08 was found instead of *m*/*z* 175.02, which indicated that the glucuronic acid was bound to the glucose group of M14.

M3 at 1.69 min and M16 at 3.04 min showed the same [M − H]^−^ at *m*/*z* 313.06, which also showed the highly similar fragment ions in the MS^2^ spectrum. The fragment ions at *m*/*z* 137.02 and 93.03 were consistent with the p-hydroxybenzyl acid unit. M3 and M16 were tentatively identified as glucuronide-conjugated of p-hydroxybenzyl acid.

M4 at 1.79 min showed the same precursor ion at *m*/*z* 299.0769 as M14 (2.69 min). However, M4 produced different fragment ions at *m*/*z* 123.0456 and 105.0349, which were agreement with the p-hydroxybenzyl alcohol unit. Therefore, M4 was assigned as the glucuronide conjugate of p-hydroxybenzyl alcohol.

M5 at 1.88 min with the precursor ion [M − H]^−^ at *m*/*z* 315.0715 was 16 Da higher than M4. The produced fragment ion at *m*/*z* 139.0409 was the agreement with the hydroxylate of p-hydroxybenzyl alcohol. Therefore, M5 was identified as the glucuronide conjugate of the hydroxylated p-hydroxybenzyl alcohol.

M7 at 2.16 min produced a precursor ion at *m*/*z* 393.0131. The fragment ions at *m*/*z* 93.0343, 137.0246, 216.9809 and 313.0562 were consistent with M3 and M16. M7 was tentatively identified as sulfate and glucuronide conjugated metabolites of p-hydroxybenzoic acid.

M17 at 3.24 min produced a precursor ion at *m*/*z* 297.0609. The product ion at *m*/*z* 121.0295 in the MS^2^ spectrum was consistent with the p-hydroxybenzaldehyde unit, indicating that M17 was the corresponding glucuronide conjugate.

M22 with the precursor ion at *m*/*z* 461.1292 was tentatively identified as glucuronide conjugated metabolites of gastrodin. The fragment ions at *m*/*z* 105.0335, 123.0443 and 337.0770 indicated that the glucuronide conjugated to the glucose of gastrodin.

M33, M37, M40, M48 and M53 showed the same precursor at *m*/*z* 633.20 were eluted at different times. In the MS^2^ spectra of all the five, the product ion at *m*/*z* 457.17 and its series-produced ions were agreement with those of M51 or M56. M33 and M37 showed the product ion at *m*/*z* 285.10 instead of the fragments at *m*/*z* 337 and 461. Therefore, the glucuronic acid of M33 and M37 were conjugated with the alcohol hydroxyl of the gymnoside I (II). M40, M48 and M53 showed the product ions at *m*/*z* 189.08, 337.08 and 461.13 indicated a glucuronic acid conjugated to the glucose of gymnoside I (II).

M52 at 12.72 min and M55 at 12.98 min were a pair of isomers. The precursor ion at *m*/*z* 471.15 was 14 Da higher than gymnoside I (II). The product ion at *m*/*z* 299.08 demonstrated the oxidation of the glucosyl, which had the same fragments as M4. So, the structures of M52 and M55 were formed by conjugating the glucuronic acid to the aglycon of gymnoside I (II).

M61 and M66 showed the same precursor ion at *m*/*z* 739.25, which was 14 Da higher than that of militarine. Similar to M52 and M55, the characteristic fragment ions at *m*/*z* 175.02, 299.08 and 471.15 also suggested that M61 and M66 were the glucuronide conjugates of the aglycon of militarine.

#### 2.3.7. Esterification Metabolites

M67, M68 and M69 were eluted at 17.74, 18.04 and 18.17 min, respectively. The [M − H]^−^ ion of all these compounds was observed at *m*/*z* 791.30, calculated as the same molecular formula of C_35_H_52_O_20_. The MS^2^ spectra showed the characteristic fragment ions of glycosylated gymnoside I(II) (M34 and M36 at *m*/*z* 619.22) and isobutyl-malic acid (M32 at *m*/*z* 189.08). By comparing their mass weights, the metabolites could be the esterification products (791 = 620 + 190 − 18 − 1) of both. Therefore, M67, M68 and M69 were identified as the esterification products formed by reacting the glycosylated gymnoside I (II) with isobutyl-malic acid.

M70 (18.18 min) and M71 (18.35 min) showed the same [M − H]^−^ ion at *m*/*z* 629.25 with the same molecular formula of C_29_H_42_O_15_, which was 162 Da (glycosyl moiety) less than M67–M69. In the MS^2^ spectrum, characteristic fragment ions of gymnoside I (II) (M51 and M56 at *m*/*z* 457.17) and isobutyl-malic acid (M32 at *m*/*z* 189.08) showed the same esterification pattern above (629 = 458 + 190 − 18 − 1). M70 and M71 were identified as the esters formed by reacting gymnoside I (II) with isobutyl-malic acid.

### 2.4. Distribution and Excretion of Militarine and Its Metabolites in Biosamples

The distribution patterns and relative contents of militarine and its metabolites in different biological samples were exhibited in Figure 5, and the corresponding extracted ion chromatograms (XICs) were presented in Figure 6 and Figure 7. The dosing militarine was detected in all samples except for feces, brain and CSF. Moreover, the hydrolysates, including isobutyl-malic acid (M32), gymnoside II (M51) and gymnoside I (M56), were detected in all samples except the brain and CSF. While gastrodin (M12) was detected in all samples except feces. At the same time, p-hydroxybenzyl alcohol glucuronide (M4), p-hydroxybenzyl alcohol sulfate (M11), gastrodin-COOH (M14) and p-hydroxybenzoic acid sulfate (M20) were detected in all samples.

In rat plasma, 20 metabolites were detected, including eight phase I metabolites and 12 phase II metabolites. Among them, the hydrolysates of militarine, including isobutyl-malic acid, gymnoside I and gymnoside II reached higher concentrations with the relative contents above 9%. In rat urine, 49 metabolites, including 11 phase I and 38 phase II, were detected. Two hydrolysates (isobutyl-malic acid and gymnoside I) and two sulfates (p-hydroxybenzyl alcohol sulfate and p-hydroxybenzoic acid sulfate) were the major metabolites, with the relative content exceeding 10%. All 20 metabolites in plasma were present in the urine, but at higher concentrations, which suggested that militarine and its metabolites underwent biotransformation and excretion mainly in the kidney.

In bile, 33 metabolites were detected. The sum of the relative contents of militarine, gymnoside I and gymnoside II reached 80%, which indicated that militarine could hydrolyze to gymnoside I and gymnoside II in the liver and excreted through bile. High levels of these three components in bile might be reabsorbed back into the blood and achieve the hepatoenteral circulation. Fifty-two metabolites were detected in feces samples, among which the relative content of isobutyl-malic acid reached 77%, and the other metabolites were less than 5%, which indicated that isobutyl-malic acid was excreted mainly through feces. In addition, there were two kinds of special metabolites in feces: (a) Glycosylated gymnoside I (II) with numerous isomers including the monosaccharide, disaccharide and trisaccharide conjugates; (b) esterification metabolites (M67–M71).

In CSF and brain, six metabolites, including p-hydroxybenzaldehyde (M28), gastrodin (M12), gastrodin-COOH (M14), sulfate conjugated p-hydroxybenzoic acid (M20), p-hydroxybenzyl alcohol sulfate conjugate (M11) and glucuronic acid conjugate (M4), were detected. To confirm the metabolites derived from gastrodin (primary metabolite of militarine) above, we also studied the metabolites in CSF after oral administration of gastrodin as the parallel experiment. All six metabolites detected in CSF after oral administration of militarine were highly consistent with those of gastrodin treated rats (Figure 6E,F). The result was different from the previous reports of the metabolism of gastrodin, which only found gastrodin and its aglycone in the brain tissue or CSF [29,30]. The five new metabolites found in CSF were worth further studying, especially for the glucuronic acid and sulfuric acid conjugates, which accounted for 66% in the total relative content. The conjugates were hydrophilic compounds with poor BBB permeability, which were difficult to get into the brain. Thus, the conjugates might be mainly formed within the brain by the glucuronosyltransferase and sulfotransferases [31,32]. The metabolites found in CSF of this study provide a better reference to elucidate the mechanism of militarine in the claimed neuropharmacological effects.

Thirty-three, 26, 25, 14, 13, and 12 metabolites were detected in the kidney, duodenum, liver, heart, lung and spleen, respectively. Among them, gastrodin-COOH (M14), gymnoside II (M51) and gymnoside I (M56) were predominated in the kidney, liver, lung, spleen, heart, and duodenum, with the relative contents above 9%. The relative content of militarine, M51 and M56 were 11%, 26% and 25% in the duodenum, respectively, which suggested the duodenum was primary absorption and metabolic tissue of militarine. The relatively high content of M14 (>31%) indicated that gastrodin underwent the oxidation in the liver.

### 2.5. Metabolic Profiles of Militarine in Rats

In the present study, a total of 71 metabolites were detected and identified in rats after oral administration of militarine, which indicated that militarine proceed extensive biological metabolism after oral administration. The proposed metabolic profiles of militarine in rats were shown in Figure 5. Both phases I and II reactions were involved in militarine metabolism, whereas hydrolysis was the dominant metabolic pathway. As a common ester, militarine was converted to gymnoside I and II, gastrodin, and isobutyl-malic acid by carboxylesterase [33]. Then, these primary hydrolytic metabolites could proceed further biological metabolism include hydrolyzation, oxidation, and conjugation. Otherwise, the oxidation, glycosylation and glucuronidation products of the intact militarine were also detected. Structurally, the tentatively identified metabolites could be categorized into four major groups: (a) Metabolites with the entire molecule of militarine (M33, M37, M61, M64–M66); (b) metabolites with gymnoside I (II) (M33, M37, M61, M64–M66); (c) metabolites with isobutyl-malic acid (M29, M30, M31); (d) metabolites with gastrodin (M1-M28).

The metabolites, whose chemical structure containing gymnoside I, gymnoside II or gastrodin, were about 91% in the total number of metabolites. Moreover, there were 27 metabolites derived from gastrodin, and six of them were detected in the brain and CSF. Militarine and its first hydrolytic metabolites (M51, M56) all have the potential to produce gastrodin. It had been reported that militarine has effects on CNS by increasing the SOD (superoxide dismutase) activity and GSH (glutathione) level [20], decreasing the Bax (BCL2 associated X protein) and caspase-3 expression [4], and preventing the amyloid-beta-induced cytotoxicity [34]. Coincidentally, many studies clarified that gastrodin exerted excellent effects of the CNS activities, such as antidepressant, anxiolytic, and learning and memory improvements, and the underlying mechanisms may be involved in SOD, GSH, Bax, and caspase-3 [35]. At the same time, such a higher relative content of gastrodin in CSF was 21.78%, which indicated it might play a predominant role in the CNS therapeutic effects of militarine. These results suggested that militarine acted as a prodrug of gastrodin in rats, and the metabolites found in CSF might be the bioactive components responsible for the pharmacological features of militarine on CNS.

Additionally, there were about 18 metabolites derived from gastrodin in the kidney and a lot of them were at a higher concentration than other tissues, which meant the gastrodin hydrolyzed from militarine was mainly further metabolized in the kidney. It was consistent with the metabolism of gastrodin reported in the previous studies [36,37]. The extensive conjugation process appeared in the pathway, which may affect the bioactive properties of the parent compound besides gastrodin. However, the biological activities of the conjugated metabolites above were still mostly unknown; thus, the bioactivities of those metabolites were also worthy of further research.

## 3. Materials and Methods

### 3.1. Metabolic Profiles of Militarine in Rats

The reference compounds of p-hydroxybenzyl alcohol, p-hydroxybenzaldehyde, p-hydroxybenzoic acid and gastrodin were purchased from Aladdin Bio-Chem Technology Co., Ltd. (Shanghai, China). Militarine was obtained from Tauto Biotech Co., Ltd. (Shanghai, China). Gymnoside I and gymnoside II were prepared by alkaline hydrolysis from militarine and purified by semi-preparative liquid chromatography system (LC-20AR, Shimadzu, Kyoto, Japan), and the corresponding structures determined by NMR.

Formic acid and acetic acid of mass-spectrometry grade, acetonitrile and methanol of HPLC grade were obtained from Merck (Darmstadt, Germany). All aqueous solutions were prepared with ultra-pure water produced from a Milli-Q^®^ Reference System (Merck, Darmstadt, Germany). Other reagents were of analytical grade or higher if not otherwise stated.

### 3.2. Animals and Drug Administration

Male Sprague-Dawley (SD) rats (body weight: 180–220 g) were obtained from Shanghai SLAC Lab. Animal Co., Ltd. (Shanghai, China). The animals were fed with standard laboratory food and water for 7 days before the experiments. Rats were fasted for 12 h before drug administration with free access to water. Animal experiments were carried out according to the Guidelines for the Care and Use of Laboratory Animals and were approved by the Animal Ethics Committee of Shanghai Institute for Food and Drug Control (No. IACUC-SIFDC 19043). Eighteen healthy rats were randomly assigned to six groups: Five administration groups (A, B, C, D and E) and one blank group (three rats for each group). After administration, blood, bile, CSF and tissue samples were collected from groups A and B at 1 or 3 h, while urine and feces from group C. Militarine solution (10.0 mg/mL militarine in water) was intragastrically administered to rats at a dosage of 60 mg/kg for groups A, B and C. Before administration, blank urine and feces were collected from group C for 4 h, while blank blood, bile, CSF and tissue samples from the blank group. For group C, urine and fecal samples were collected for 0–24 h from rats housed in metabolic cages. Feces were weighed and homogenized in 5 vol. of ice-cold saline at 4 °C, then centrifuged at 14,000 rpm for 10 min to get the supernatant as fecal samples. After administration of militarine, rats in groups A and B were anesthetized with an intraperitoneal (i. p.) injection of chloral hydrate (500 uL, 10%) at 45 or 165 min, and bile duct intubation was performed to collect bile for 5 min (about 200 uL). Next, cerebrospinal fluid (about 100 uL) was taken from the occipital foramen at 1 or 3 h. Then, approximately 2.0 mL of blood was taken from the abdominal aorta with sodium fluoride/potassium oxalate tube (BD Vacutainer™ PLUS, Becton Dickinson, Franklin Lakes, NJ, USA), and plasma was prepared by centrifugation at 4000 rpm for 10 min. Finally, after perfusion with 250 mL ice-cold saline at 1 or 3 h, the heart, liver, spleen, lung, kidney, duodenum and brain were removed and then rinsed in ice-cold saline before gently blotting on absorbent paper. Each organ was cut into small pieces with scissors and homogenized in 5 vol. of ice-cold saline, using a tissue homogenizer. After centrifugation at 14,000 rpm for 10 min, the supernatants were obtained as tissue samples. For groups D and E, cerebrospinal fluid (100 uL) was drawn from the occipital foramen at 1 or 3 h after oral administration of gastrodin solution (10.0 mg/mL gastrodin in water 50 mg/kg). Finally, all the samples were stored at −80 °C until analysis.

### 3.3. Sample Preparation

The pre-treated and merged sample (0.5 mL) diluted with 2.0 mL of 0.1% formic acid in water was loaded onto the Oasis^®^ PRiME HLB Cartridge (3 cc/60 mg; Waters, Milford, MA, USA) on a vacuum manifold at a flow rate of 0.2 mL/min. The cartridge was washed with 4.0 mL of 0.1% formic acid in water. After purging with air, the cartridge was eluted with 3.0 mL of 5.0% ammonia in methanol at a flow rate of 0.2 mL/min. After the eluate was adjusted to pH 2.0 with 10% formic acid in methanol and evaporated at 40 °C in vacuo, the residue was dissolved in 100 μL of methanol with vortex mixing for 30 s and then centrifuged at 14,000 rpm for 10 min. The supernatant was ready for injection analysis.

### 3.4. Instrumental Parameters

Liquid chromatography was performed on an ACQUITY UPLC system (Waters, Milford, MA, USA) equipped with a binary pump with on-line degasser, auto-sampler, and column oven. Chromatographic separation was performed on an ACQUITY BEH C18 column (2.1 × 100 mm, 1.7 μm, Waters, Milford, MA, USA) with the temperature at 40 °C. The mobile phase consisted of water containing 0.1% acetic acid (A) and acetonitrile containing 0.5% formic acid (B). The analytes were eluted by the following linear gradient program—1.0% B at 0–2.0 min; 1.0% B→25% B at 2–15 min; 25%→35% B at 15–24 min; 95% B at 24.1–26 min a flow rate of 0.4 mL/min. The equilibration time was 2 min. The injection volume of reference compounds and samples was 3 uL.

Mass spectrometry was performed on a quadrupole-time-of-flight mass spectrometer (X500R, AB SCIEX, Framingham, MA, USA), equipped with electrospray ionization (ESI) source. The mass spectrometer was operated in negative ion mode. The ESI source temperature was maintained at 550 °C. The mass data acquisition was set in the range of *m*/*z* 100–1500 Da. The mass spectroscopic analysis was performed under the following conditions: curtain gas was 45 psi, GAS 1 and GAS 2 were set at 50 psi; ion spray voltage was set at −4500 V; collision energy was −40 V. MS and MS^2^ data were acquired through MS scan and IDA MS^2^ scans by Sciex OS1.5 software (AB SCIEX, Framinghamcity, MA, USA, 2019) with mass defect filter and dynamic background subtraction.

## 4. Conclusions

In this work, we established a sensitive and reliable method for exploration and identification of the metabolites of militarine in rats after oral administration by UPLC-Q-TOF-MS/MS. Among 71 identified metabolites, 57 metabolites were reported firstly. The result provides a detail metabolic profile of militarine and the distribution of militarine and its metabolites in rats, which will help to reveal the chemical basis of militarine pharmacology. Militarine plays a pharmacological role through hydrolytic products, which can be further metabolized through hydrolysis, oxidation, glycosylation, sulfation and glucuronidation. In CSF, five metabolites except gastrodin found were not reported before, which was confirmed with the metabolites of the gastrodin administration in rats in our study. The result could provide more in-depth insights into the active constituents working in CNS and would be fundamental for further revealing the neuroprotective mechanism of militarine. Base on the metabolic pathways of militarine in rats, we can speculate the similar compounds with glucosyloxybenzyl ester would follow the same biotransformation process in vivo after oral administration to produce active metabolites (such as gastrodin and its further metabolites), which exert the nervous protective effects.

## Figures and Tables

**Figure 1 molecules-25-01082-f001:**
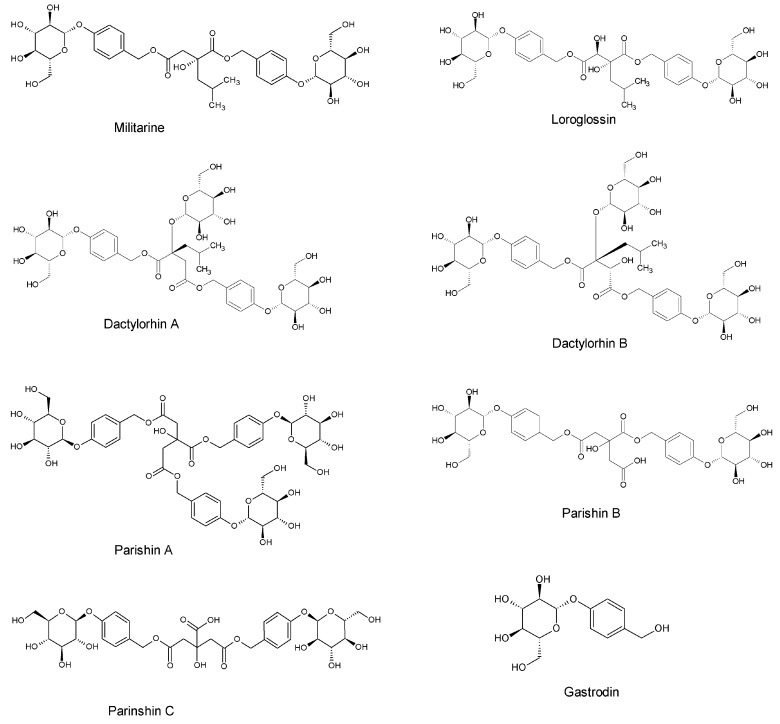
The structures of militarine, gastrodin and other glucosylated benzyl esters.

**Figure 2 molecules-25-01082-f002:**
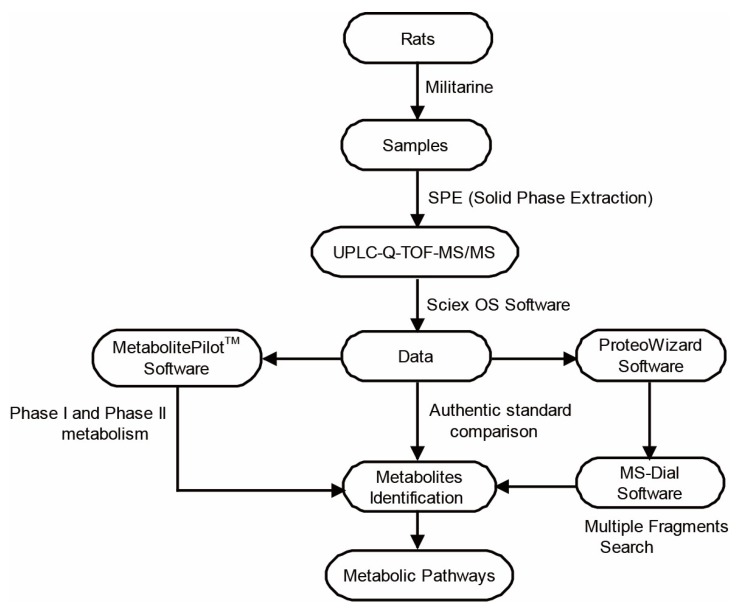
Summary diagram of systematic screening and identification of metabolites in multi-samples with the developed strategy and methodology.

**Figure 3 molecules-25-01082-f003:**
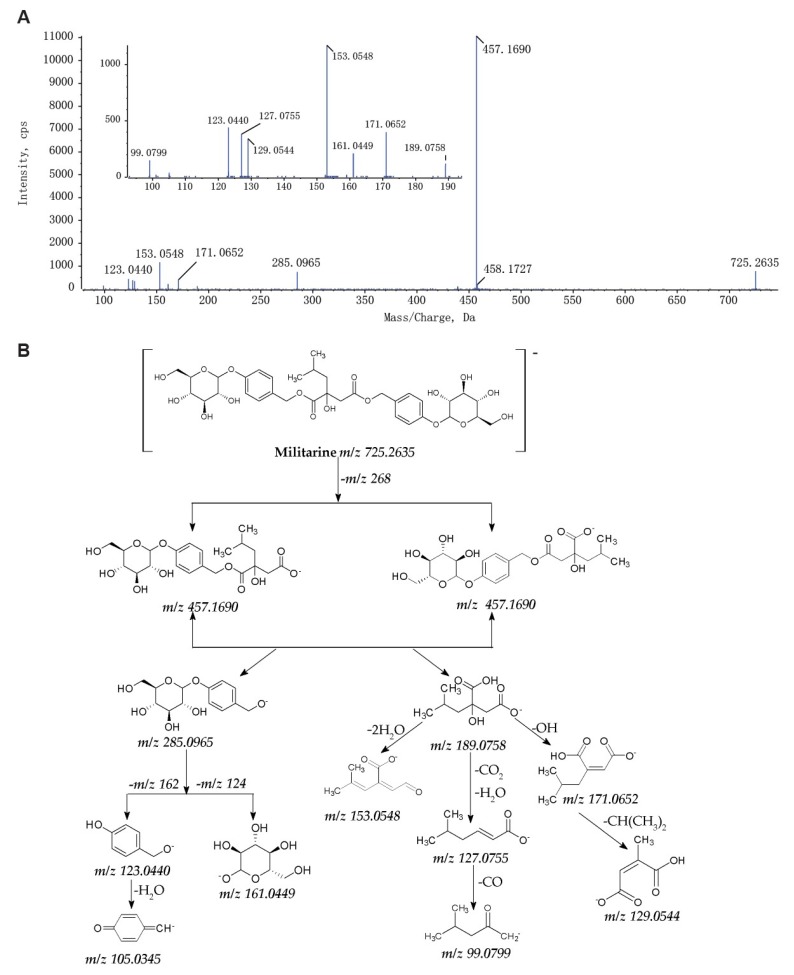
The MS^2^ spectrum (**A**) and characterized fragmentation pathway (**B**) of militarine.

**Figure 4 molecules-25-01082-f004:**
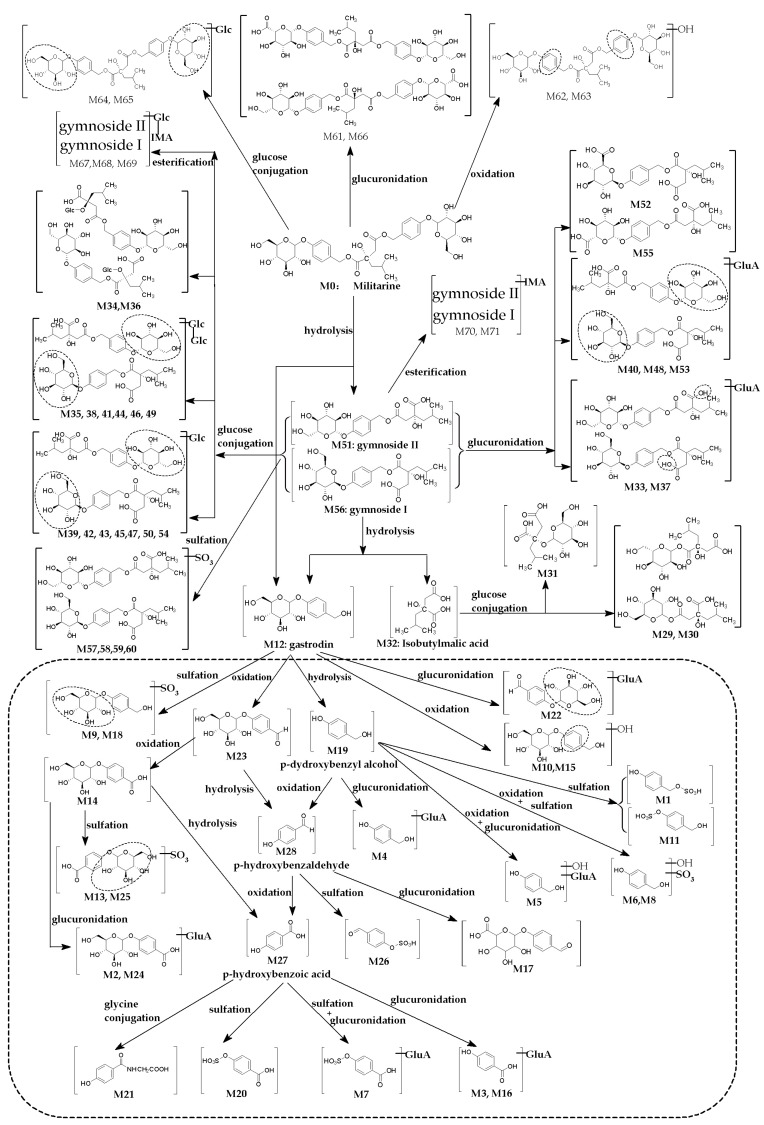
The proposed chemical structures of the metabolites and identified metabolic pathways.

**Figure 5 molecules-25-01082-f005:**
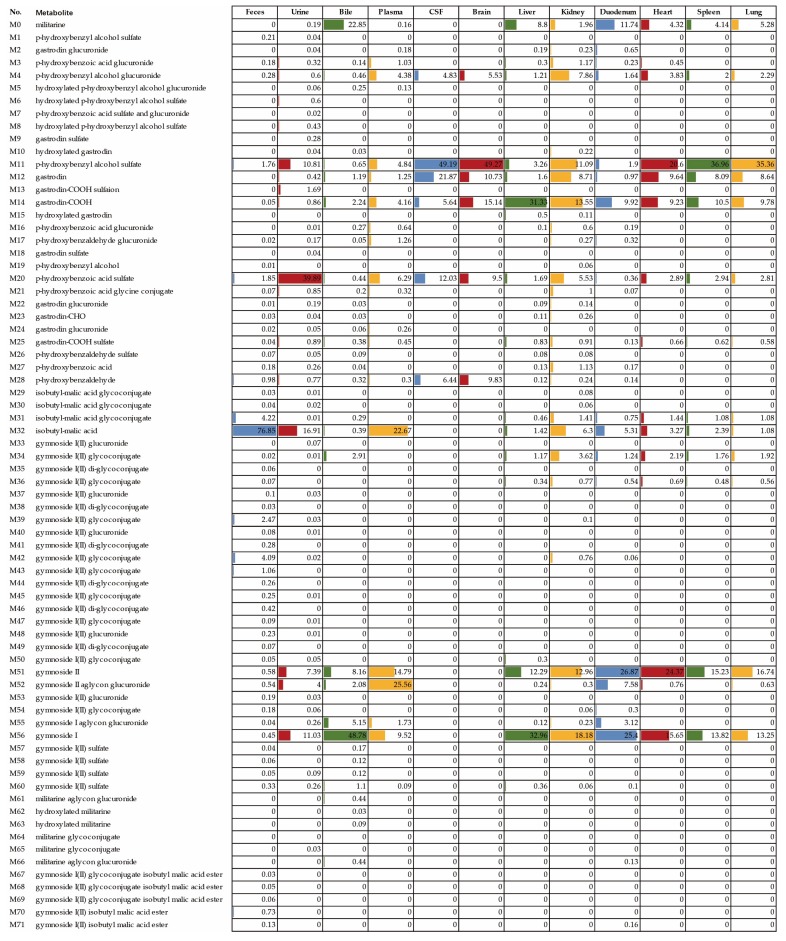
The distribution of 71 metabolites of militarine and their relative contents in rat feces, urine, bile, plasma, CSF and various organs. The relative content of the metabolites in each sample was calculated by (peak area of a metabolite/total peak areas of all detected metabolites) × 100%.

**Figure 6 molecules-25-01082-f006:**
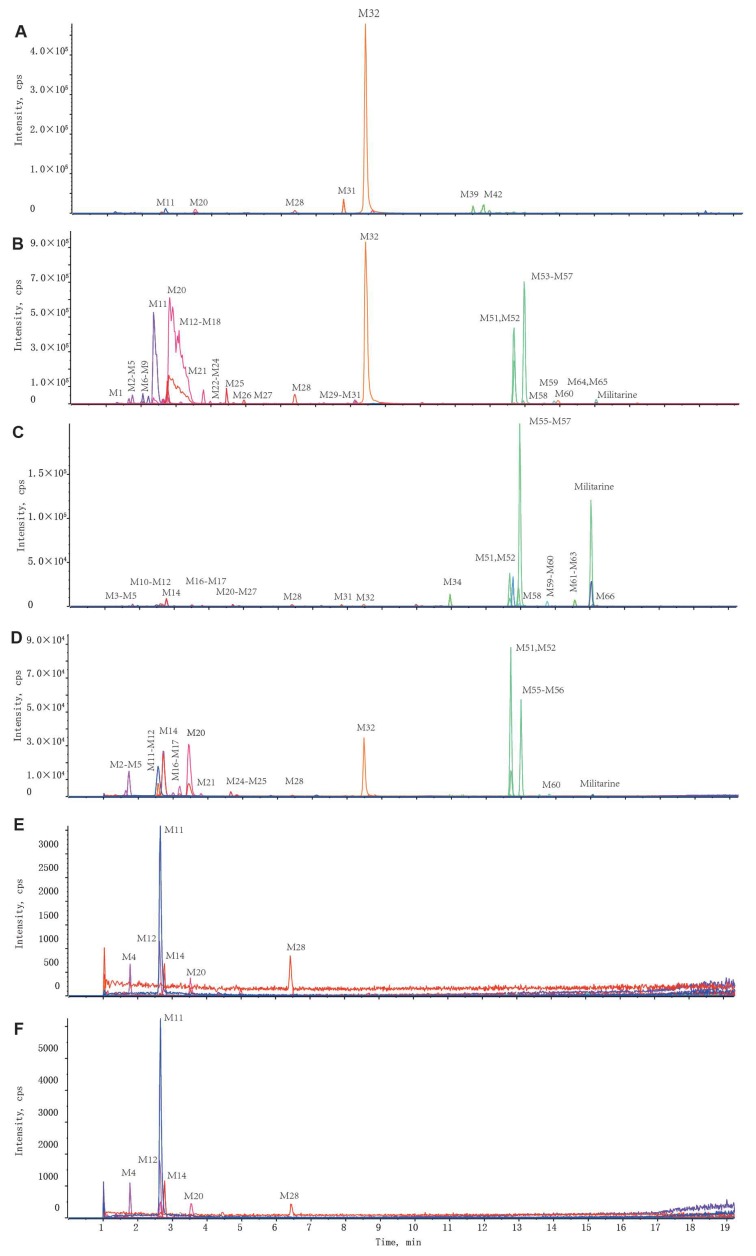
Representative extracted ion chromatograms (XICs) of the metabolites of militarine in biological samples of rats. (**A**) Feces; (**B**) urine; (**C**) bile; (**D**) plasma; (**E**) CSF; (**F**) CSF (gastrodin administration).

**Figure 7 molecules-25-01082-f007:**
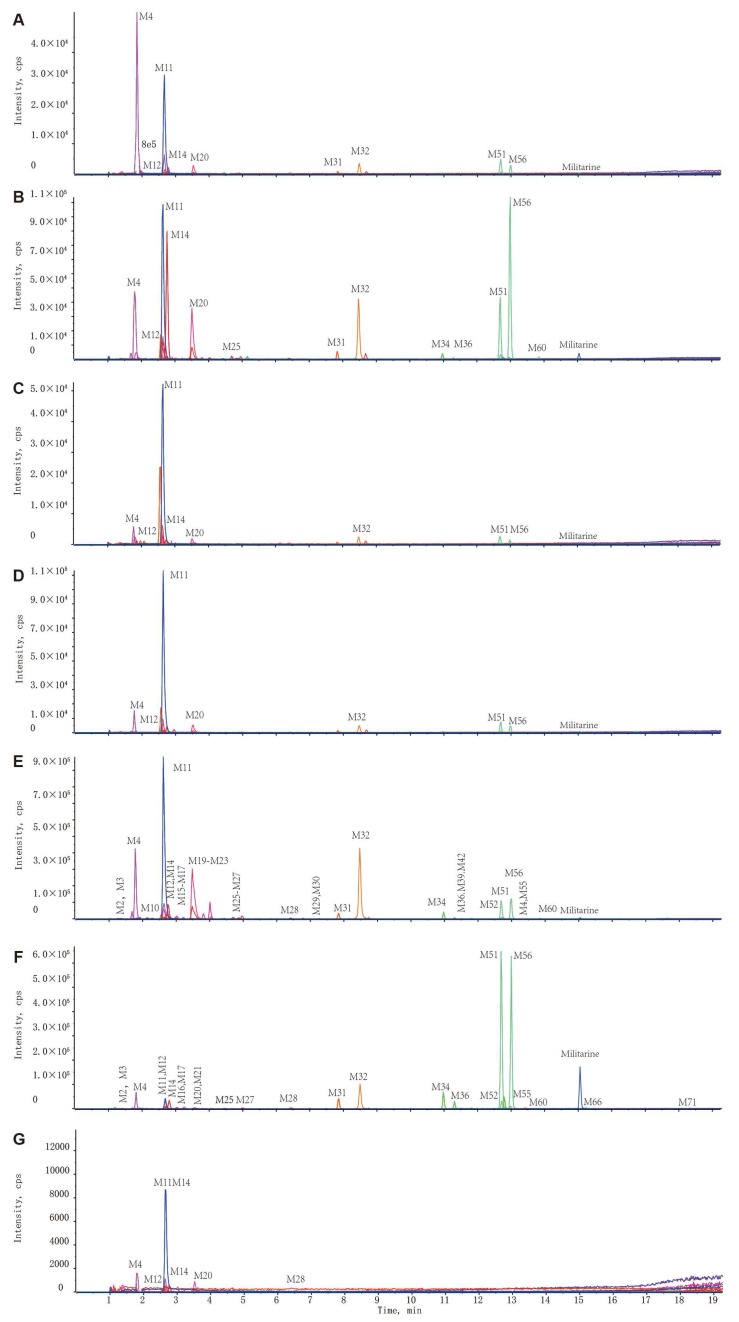
Representative extracted ion chromatograms (XICs) of the metabolites of militarine in tissue samples. (**A**) Heart; (**B**) liver; (**C**) spleen; (**D**) lung; (**E**) kidney; (**F**) duodenum; (**G**) brain.

**Table 1 molecules-25-01082-t001:** The metabolites of militarine in rats after oral administration.

No.	R.T. (min)	*m*/*z*	Error (ppm)	Adduct/Charge	Formula	Product Ions	Metabolites
M0 *	15.06	771.2703	−1.8	[M + FA − H]^−^	C_34_H_46_O_17_	99, 105, 123, 127, 129, 153, 189,285, 439, 457	/
M1	1.38	203.0021	0.7	[M − H]^−^	C_7_H_8_O_5_S	77, 80, 105, 123	p-hydroxybenzyl alcohol sulfate
M2	1.68	475.1089	−0.9	[M − H]^−^	C_19_H_24_O_14_	93, 137, 175, 299	gastrodin glucuronide
M3	1.69	313.0571	0.7	[M − H]^−^	C_13_H_14_O_9_	93, 137, 175	p-hydroxybenzoic acid glucuronide
M4	1.79	299.0769	−1.3	[M − H]^−^	C_13_H_16_O_8_	105, 123, 175	p-hydroxybenzyl alcohol glucuronide
M5	1.88	315.0715	−2.0	[M − H]^−^	C_13_H_16_O_9_	139, 175	hydroxylated p-hydroxybenzyl alcohol glucuronide
M6	2.12	218.9967	−0.9	[M − H]^−^	C_7_H_8_O_6_S	80, 93, 97, 121, 139	hydroxylated p-hydroxybenzyl alcohol sulfate
M7	2.16	393.0131	−0.6	[M − H]^−^	C_13_H_14_O_12_S	93, 97, 137, 175, 193, 217, 313	p-hydroxybenzoic acid sulfate and glucuronide
M8	2.26	218.9967	−0.9	[M − H]^−^	C_7_H_8_O_6_S	80, 93, 97, 121, 139	hydroxylated p-hydroxybenzyl alcohol sulfate
M9	2.61	365.0547	−0.2	[M − H]^−^	C_13_H_18_O_10_S	80, 97, 241	gastrodin sulfate
M10	2.64	301.0924	−0.6	[M − H]^−^	C_13_H_18_O_8_	93, 123, 139	hydroxylated gastrodin
M11	2.66	203.0018	−0.6	[M − H]^−^	C_7_H_8_O_5_S	80, 97, 105, 123	p-hydroxybenzyl alcohol sulfate
M12 *	2.67	345.1189	−0.6	[M + AcO − H]^−^	C_13_H_18_O_7_	105, 123	gastrodin
M13	2.70	379.0345	−1.1	[M − H]^−^	C_13_H_16_O_11_S	80, 93, 97, 137, 241, 299	gastrodin-COOH sulfaion
M14	2.79	299.0771	−0.5	[M − H]^−^	C_13_H_16_O_8_	93, 137	gastrodin-COOH
M15	2.81	301.0925	−1.3	[M − H]^−^	C_13_H_18_O_8_	93, 123, 139	hydroxylated gastrodin
M16	3.04	313.0559	−1.8	[M − H]^−^	C_13_H_14_O_9_	93, 137, 175, 193	p-hydroxybenzoic acid glucuronide
M17	3.24	297.0609	−2.2	[M − H]^−^	C_13_H_14_O_8_	121, 175	p-hydroxybenzaldehyde glucuronide
M18	3.32	365.0546	−0.4	[M − H]^−^	C_13_H_18_O_10_S	80, 97, 241	gastrodin sulfate
M19 *	3.43	123.0451	−0.4	[M − H]^−^	C_7_H_8_O_2_	77, 105	p-hydroxybenzyl alcohol
M20	3.53	216.9812	0.0	[M − H]^−^	C_7_H_6_O_6_S	93, 97, 137	p-hydroxybenzoic acid sulfate
M21	3.83	194.0458	−1.9	[M − H]^−^	C_9_H_9_NO_4_	93, 100, 150	p-hydroxybenzoic acid glycine conjugate
M22	4.05	461.1292	−1.9	[M − H]^−^	C_19_H_26_O_13_	105, 123, 175, 337	gastrodin glucuronide
M23	4.41	329.0878	0.1	[M + FA − H]^−^	C_13_H_16_O_7_	92, 121	gastrodin-CHO
M24	4.42	475.1093	0.0	[M − H]^−^	C_19_H_24_O_14_	93, 137, 175, 337	gastrodin glucuronide
M25	4.68	379.0338	−1.7	[M − H]^−^	C_13_H_16_O_11_S	80, 97, 241, 299	gastrodin-COOH sulfate
M26	4.80	200.9862	−0.6	[M − H]^−^	C_7_H_6_O_5_S	92, 121	p-hydroxybenzaldehyde sulfate
M27 *	4.98	137.0244	0.5	[M − H]^−^	C_7_H_6_O_3_	93	p-hydroxybenzoic acid
M28 *	6.43	121.0295	0.1	[M − H]^−^	C_7_H_6_O_2_	92	p-hydroxybenzaldehyde
M29	7.12	351.1294	−0.8	[M − H]^−^	C_14_H_24_O_10_	127, 129, 153, 189	isobutyl-malic acid glycoconjugate
M30	7.42	351.1292	−1.4	[M − H]^−^	C_14_H_24_O_10_	127, 129, 153, 189	isobutyl-malic acid glycoconjugate
M31	7.84	351.1291	−1.5	[M − H]^−^	C_14_H_24_O_10_	89, 127, 171, 307	isobutyl-malic acid glycoconjugate
M32	8.48	189.0766	0.9	[M − H]^−^	C_8_H_14_O_5_	99, 127, 129, 171	isobutyl-malic acid
M33	10.65	633.2033	−0.5	[M − H]^−^	C_27_H_38_O_17_	99, 123, 127, 129, 153, 175, 285, 457	gymnoside I(II) glucuronide
M34	10.98	619.2234	−1.5	[M − H]^−^	C_27_H_40_O_16_	123, 127, 153, 285, 351, 439	gymnoside I(II) glycoconjugate
M35	11.03	781.2778	0.8	[M − H]^−^	C_33_H_50_O_21_	129, 171, 323, 485, 609	gymnoside I(II) di-glycoconjugate
M36	11.30	619.2232	−1.9	[M − H]^−^	C_27_H_40_O_16_	153, 179, 351, 439	gymnoside I(II) glycoconjugate
M37	11.33	633.2039	0.4	[M − H]^−^	C_27_H_38_O_17_	123, 127, 153, 175, 193, 285, 457	gymnoside I(II) glucuronide
M38	11.34	781.2772	0.0	[M − H]^−^	C_33_H_50_O_21_	129, 323, 485, 609	gymnoside I(II) di-glycoconjugate
M39	11.53	619.2233	−1.7	[M − H]^−^	C_27_H_40_O_16_	99, 127, 129, 323	gymnoside I(II) glycoconjugate
M40	11.56	633.2035	−0.2	[M − H]^−^	C_27_H_38_O_17_	99, 123, 127, 129, 189, 337, 457, 461	gymnoside I(II) glucuronide
M41	11.70	781.2772	0.0	[M − H]^−^	C_33_H_50_O_21_	323, 485, 609	gymnoside I(II) di-glycoconjugate
M42	11.82	619.2232	−1.8	[M − H]^−^	C_27_H_40_O_16_	99, 127, 129, 153, 179, 323, 447	gymnoside I(II) glycoconjugate
M43	11.98	619.2237	−1.0	[M − H]^−^	C_27_H_40_O_16_	99, 123, 127, 161, 285, 323, 447	gymnoside I(II) glycoconjugate
M44	12.13	781.2767	−0.7	[M − H]^−^	C_33_H_50_O_21_	323, 485, 609	gymnoside I(II) di-glycoconjugate
M45	12.29	619.2234	−1.5	[M − H]^−^	C_27_H_40_O_16_	99, 127, 129, 323, 447	gymnoside I(II) glycoconjugate
M46	12.36	781.2769	−0.3	[M − H]^−^	C_33_H_50_O_21_	129, 323, 485, 609, 763	gymnoside I(II) di-glycoconjugate
M47	12.49	619.2234	−1.6	[M − H]^−^	C_27_H_40_O_16_	99, 127, 129, 323, 447	gymnoside I(II) glycoconjugate
M48	12.67	633.2042	0.9	[M − H]^−^	C_27_H_38_O_17_	123, 127, 129, 153, 189, 337, 457, 461	gymnoside I(II) glucuronide
M49	12.68	781.2769	−0.4	[M − H]^−^	C_33_H_50_O_21_	179, 323, 485, 609, 763	gymnoside I(II) di-glycoconjugate
M50	12.69	619.2239	−0.7	[M − H]^−^	C_27_H_40_O_16_	99, 127, 129, 189, 323, 447	gymnoside I(II) glycoconjugate
M51 *	12.71	457.1701	−3.2	[M − H]^−^	C_21_H_30_O_11_	105, 123, 127, 129, 153, 189, 285, 439	gymnoside II
M52	12.72	471.1511	0.7	[M − H] ^−^	C_21_H_28_O_12_	99, 123, 127, 129, 175, 189, 299	gymnoside II aglycon glucuronide
M53	12.94	633.2038	0.3	[M − H]^−^	C_27_H_38_O_17_	99, 123, 129, 189, 337, 457, 461	gymnoside I(II) glucuronide
M54	12.96	619.2230	−2.3	[M − H]^−^	C_27_H_40_O_16_	99, 127, 129, 323, 447	gymnoside I(II) glycoconjugate
M55	12.98	471.1509	0.3	[M − H]^−^	C_21_H_28_O_12_	99, 127, 129, 175, 189, 299	gymnoside I aglycon glucuronide
M56 *	13.00	457.1721	1.3	[M − H] ^−^	C_21_H_30_O_11_	99, 105, 123, 127, 129, 153, 189, 285	gymnoside I
M57	13.07	537.1281	−0.4	[M − H]^−^	C_21_H_30_O_14_S	97, 129, 189, 241, 457, 491	gymnoside I(II) sulfate
M58	13.37	537.1286	0.4	[M − H]^−^	C_21_H_30_O_14_S	97, 189, 241, 241	gymnoside I(II) sulfate
M59	13.55	537.1284	−1.8	[M − H]^−^	C_21_H_30_O_14_S	97, 129, 189, 241, 457, 491	gymnoside I(II) sulfate
M60	13.83	537.1278	−1.0	[M − H]^−^	C_21_H_30_O_14_S	97, 127, 129, 153, 189, 241	gymnoside I(II) sulfate
M61	14.62	739.2453	−3.0	[M − H]^−^	C_34_H_44_O_18_	123, 175, 189, 299, 471	militarine aglycon glucuronide
M62	14.66	741.2611	0.0	[M − H]^−^	C_34_H_46_O_18_	123, 129, 139, 189, 285, 301, 457, 473	hydroxylated militarine
M63	14.75	741.2608	−0.5	[M − H]^−^	C_34_H_46_O_18_	123, 129, 139, 189, 285, 301, 457, 473	hydroxylated militarine
M64	14.79	887.3178	−1.5	[M − H]^−^	C_40_H_56_O_22_	153, 323, 447, 457, 619	militarine glycoconjugate
M65	14.92	887.3182	−1.0	[M − H]^−^	C_40_H_56_O_22_	129, 153, 323, 447, 457, 619	militarine glycoconjugate
M66	15.12	739.2448	−2.1	[M − H]^−^	C_34_H_44_O_18_	129, 175, 189, 457, 471	militarine aglycon glucuronide
M67	17.74	791.2985	0.7	[M − H]^−^	C_35_H_52_O_20_	129, 189, 323, 447, 619, 661	gymnoside I(II) glycoconjugate isobutyl malic acid ester
M68	18.04	791.2956	−3.0	[M − H]^−^	C_35_H_52_O_20_	129, 323, 447, 498, 619	gymnoside I(II) glycoconjugate isobutyl malic acid ester
M69	18.17	791.2977	−0.2	[M − H]^−^	C_35_H_52_O_20_	129, 189, 323, 447, 619	gymnoside I(II) glycoconjugate isobutyl malic acid ester
M70	18.18	629.2450	−0.2	[M − H]^−^	C_29_H_42_O_15_	99, 123, 127, 129, 153, 189, 285, 457	gymnoside I(II) isobutyl malic acid ester
M71	18.35	629.2440	−1.7	[M − H]^−^	C_29_H_42_O_15_	105, 123, 127, 129, 189, 285, 457	gymnoside I(II) isobutyl malic acid ester

* Metabolites marked with (*) were absolutely identified by reference standard. R.T.: Retention time; FA: Formic acid; AcO: Acetic acid.

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
