# Peer review of "Metabolite Profiling and Distribution of Militarine in Rats Using UPLC-Q-TOF-MS/MS"

_molecules, 2020, doi:10.3390/molecules25051082_

Round 1

Reviewer 1 Report

The manuscript “Metabolite Profiling and Distribution of Militarine in Rats using UPLC-Q-TOF-MS/MS” aimed to investigate in vivo the metabolite profile and the distribution of militarine a natural glucosyloxybenzyl 2-isobutylmalate, isolated from Bletilla striata. A total of 71 metabolites with 57 new metabolites were identified in rats by ultra-high-performance liquid chromatography combined with quadrupole time-of-flight tandem mass spectrometry (UPLC-Q-TOF-MS/MS). The paper shows for the first time an extended analysis of Militarine metabolites extracted from different biological fluids in vivo. The paper is clear in the presentation of the data collected and the experimental design is overall well articulated; the possibility to investigate the metabolites in different fluids enriches the work and provides the basis for understanding the biological action of militarine. The characteristics of the rats used in the experiments, the total number of animals and the volumes of biological fluids collected are not adequately reported, therefore the robustness of the data presented is unclear.

The number of metabolites identify is certainly high but some more detailed comments are missing that can be traced back to the pharmacological action of militarine.

Reviewer 2 Report

The manuscript by Li et al describe the structures of 71 metabolites of Militarine and their relative distribution across biofluids and tissues in rats. The work is extremely well described with ample evidence to support the results and conclusions made by the authors. It is an enjoyable paper to read with a lot of information and provides an excellent lesson for mass spectral identification of metabolites. I have some very minor comments, however I believe this work is of very high quality and should appeal to the readers of molecules.

Introduction

Page 2 line 54: What are these 4 metabolite previously identified?

2.1 method optimisation

Page 3 line 72-73: The short run time of what? Its not really clear what is being described here

Page 3 line 82: “right responds” this needs to be rephrased

2.2 The strategy for identification of metabolites

Page 4 line 103, The city and country are missing for Sciex details

Page 6, line 186: The authors say natural glucose, I’ve not seen this termed use before, does the natural need to be included?

2.4. Distribution and excretion of militarine and its Metabolites in Biosamples

Page 12 line 312: Prototype militarine is poorly phrased, parent, dosing or unmetabolised militarine or something along those lines would be a better way to describe this.

Page 12, lines 344 – 345: “Glucoronide and sulfage cnjugates were hydrophilic compounds difficulty crossing the BBB. This needs to be rephrased as the sentence currently doesn’t make sense. Also are the authors suggesting these metabolites are formed in the CNS or cross into the CSF?

Figure 5: Some of the text here is difficult to read

3.4 Instrument parameters

Page 17: were the ESI voltages the same in both modes?

Reviewer 3 Report

I found the manuscript interesting, well organized and with a very complete and coherent experimental work. I have not detected errors in the manuscript. Consequently, I recommend its publication in Molecules.

As advice, I recommend rewriting two paragraphs in the Introduction and Results and Discussion sections because they are confusing.

Lines 39-40 "The previous study found militarine can transform into gastrodin in rats or by liver microsomes,..."

Line 76 "... of 0.1% acetic acid and acetonitrile with 0.5% formic acid was selected,..."
